# Live-Cell Cardiac-Specific High-Throughput Screening Platform for Drug-Like Molecules That Enhance Ca^2+^ Transport

**DOI:** 10.3390/cells9051170

**Published:** 2020-05-08

**Authors:** Tory M. Schaaf, Evan Kleinboehl, Samantha L. Yuen, Lauren N. Roelike, Bengt Svensson, Andrew R. Thompson, Razvan L. Cornea, David D. Thomas

**Affiliations:** Department of Biochemistry, Molecular Biology and Biophysics, University of Minnesota, Minneapolis, MS 55455, USA; tms@ddt.umn.edu (T.M.S.); evan@ddt.umn.edu (E.K.); sly@ddt.umn.edu (S.L.Y.); roel0047@umn.edu (L.N.R.); bsven@ddt.umn.edu (B.S.); at@ddt.umn.edu (A.R.T); rc@ddt.umn.edu (R.L.C.)

**Keywords:** time-resolved FRET, biosensor, drug screening, SERCA2a, small-molecule, HTS, plate reader, fluorescence, fluorescent proteins, PIE, FLIM, fluorescence lifetime

## Abstract

We engineered a concatenated fluorescent biosensor and dual-wavelength fluorescence lifetime (FLT) detection, to perform high-throughput screening (HTS) in living cells for discovery of potential heart-failure drugs. Heart failure is correlated with insufficient activity of the sarcoplasmic reticulum Ca-pump (SERCA2a), often due to excessive inhibition by phospholamban (PLB), a small transmembrane protein. We sought to discover small molecules that restore SERCA2a activity by disrupting this inhibitory interaction between PLB and SERCA2a. Our approach was to fluorescently tag the two proteins and measure fluorescence resonance energy transfer (FRET) to detect changes in binding or structure of the complex. To optimize sensitivity to these changes, we engineered a biosensor that concatenates the two fluorescently labeled proteins on a single polypeptide chain. This SERCA2a-PLB FRET biosensor construct is functionally active and effective for HTS. By implementing 2-wavelength FLT detection at extremely high speed during primary HTS, we culled fluorescent compounds as false-positive Hits. In pilot screens, we identified Hits that alter the SERCA2a-PLB interaction, and a newly developed secondary calcium uptake assay revealed both activators and inhibitors of Ca-transport. We are implementing this approach for large-scale screens to discover new drug-like modulators of SERCA2a-PLB interactions for heart failure therapeutic development.

## 1. Introduction

The small regulatory protein phospholamban (PLB) is essential for controlling cardiac contraction through inhibition of the sarco/endoplasmic reticulum calcium ATPase (SERCA2a) [1]. Alteration in SERCA2a/PLB expression levels or PLB phosphorylation tune SERCA2a activity, and consequently the rate of myocyte relaxation and priming for the next contraction (heartbeat). A key example of tuning SERCA2a regulation is during the flight-or-fight response, in which the β-adrenergic cascade activates protein kinase A (PKA), resulting in PLB phosphorylation and relief of PLB-mediated SERCA2a inhibition. The resulting activation of SERCA2a enhances pumping of Ca^2+^ back into the lumen of the sarcoplasmic reticulum (SR). This allows relaxation of the myocyte and prepares it for Ca^2+^ release through the ryanodine receptor to initiate the next round of cardiac contraction [2]. It is well established that SERCA2a insufficiency contributes significantly to the progression of cardiac disease [3]. There have been multiple approaches to discover therapeutics that activate SERCA2a. The most promising strategy, to date, involves gene therapy to enhance SERCA2a expression in the heart through injection of adeno-associated virus carrying the SERCA2a gene [4]. However, recent clinical trials using this approach were unsuccessful, probably due to current limitations of adeno-associated virus (AAV)-based gene delivery [5]. Specifically, SERCA2a is a large protein near the size limit for AAV-induced overexpression, and humans generate neutralizing antibodies against the viral vector carrying SERCA2a, making repeated treatments unfeasible [6]. To overcome these challenges, we aimed to discover small molecules that activate endogenous SERCA2a. We described previously our ongoing work to develop small molecules that activate SERCA2a directly by relieving its inhibition by PLB.

We developed a structure-based drug discovery platform to find compounds that affect the structure of the SERCA2a-PLB complex in living cells, as detected by fluorescence resonance energy transfer (FRET, also known as Förster resonance energy transfer), between fluorescent proteins fused to SERCA2a and PLB [7,8]. This platform uses novel instrumentation for high-precision, high-throughput subnanosecond time-resolved fluorescence lifetime (FLT) measurements in a 1536-well microplate-reader [9], applied to high-throughput screening (HTS) of small-molecule chemical libraries. The fusion of green and red fluorescent proteins (GFP/RFP) to specific locations on SERCA2a and PLB allows detection of structural rearrangements related to Ca-transport activity. When the donor (GFP) is in close proximity (<10 nm) to the acceptor (RFP), energy transfer from the donor to acceptor is likely [10], with an extremely sensitive distance (r) dependence, proportional to 1/r^6^. This would typically be monitored as a decrease in donor fluorescence intensity using a conventional plate reader (PR) or microscopy. However, FLT (time-resolved) measurements offer major advantages over standard intensity (steady-state) measurements, since FLT captures an array of time-dependent intensity points in a single measurement with nanosecond resolution, in order to determine the change in FLT due to FRET [8,9]. This results in a much greater information content and substantial improvement in precision [7,8]. Previous single-molecule FRET, fluorescence lifetime imaging microscopy (FLIM), and pulsed interleaved excitation (PIE) measurements have been shown to achieve excellent precision [11,12], but their throughput is low (much less than one sample per second) so they are not applicable to high-throughput screening. We solved the throughput problem by developing plate-readers that record FLT data, combining high precision (S/N > 100) and high throughput (>10 samples per s), as needed for HTS [8]. We applied this technology platform, combining biosensor engineering and FLT measurement technology, to disease-relevant protein targets, including SERCA (heart failure, sarcopenia) [7,9], SERCA2a-PLB (heart failure) [13], myosin (heart failure, sarcopenia) [14], myosin-actin (heart failure, sarcopenia) [14,15], ryanodine receptor (heart failure, muscular dystrophy) [16,17], tau (Alzheimer’s disease) [18], TNF receptor (inflammation, arthritis) [19,20], and aurora kinase (cancer) [21,22]. In the present study, we describe two extensions of this technology.

First, we describe an advance in biosensor engineering. FRET biosensors can be intramolecular, with donor and acceptor on the same polypeptide chain, or intermolecular, with donor and acceptor on separate polypeptide chains. We previously used *intra*molecular FRET biosensors, wherein donor and acceptor fluorescent proteins are both fused to SERCA2a in live cells [7,8,9,23]. This two-color SERCA2a biosensor detects compounds that bind to SERCA2a and change its structure. In parallel, we developed an *inter*molecular FRET biosensor in live cells, to detect compounds that disrupt the inhibitory SERCA2a-PLB complex, either by dissociating PLB from SERCA2a or by changing the structure of the complex, and we used it to discover compounds that increase SERCA2a activity due to the loss of inhibition by PLB [13]. This biosensor was transiently co-expressed in HEK293 cells, which can be impractical for large-scale HTS, because it is difficult to control the relative expression levels of the two proteins and their co-localization within the same cells and membranes. It has been previously demonstrated that a concatenated SERCA2a-PLB construct, in which PLB is tethered to SERCA2a in a single polypeptide chain, can be recombinantly expressed in Sf21 insect cells as a functionally-intact fusion protein [24]. Here, we extended this approach, adding fluorescent proteins GFP and RFP, to engineer a SERCA2a-PLB intramolecular FRET biosensor concatenated on a single polypeptide chain (denoted “fusion biosensor” below). This construct includes genetically encoded fluorophores on both SERCA2a and PLB, thus maximizing sensitivity to changes in the SERCA2a-PLB interaction. By fixing the donor-acceptor ratio at 1, we reduce the range of FRET values for the control (no compound added), while improving day-to-day reproducibility. The goals of this study were focused on drug discovery, but the technology demonstrated here can also be used to answer fundamental mechanistic questions about PLB regulation of SERCA2a [24].

Second, we describe instrumentation changes that allow *simultaneous* detection of FLTs at two wavelengths in the fluorescence emission spectrum with *subnanosecond* resolution at extremely (unprecedented) high speed during primary HTS (2-channel detection). This information is used in data analysis (Section 3.4 below), to provide effective flagging of readouts from wells containing compounds with interfering fluorescence signals, thus increasing accuracy and precision during HTS by culling such false-positives from Hit selection, therefore economizing the follow-up assays [25].

## 2. Materials and Methods

### 2.1. Molecular Biology

This SERCA2a-PLB fusion construct was generated utilizing a previously developed RFP-SERCA2a construct [23]. A DNA construct was synthesized with the C-terminal region of human SERCA2a fused to a forty-seven amino acid flexible linker, fused to the coding region of GFP-PLB. The synthetic DNA sequence was then subcloned into the RFP-SERCA2a construct using BamHI and NotI restriction enzyme sites. Each biosensor construct was subcloned into a puromycin resistant expression vector. The donor-only construct was made in a similar manner.

### 2.2. Cell Culture

HEK293-6E, obtained from the nation research council Canada, cells were transfected using 293 fectin protocol (Thermo Fisher, Waltham, MA, USA). The fusion biosensor was expressed using mammalian expression vector pTT22 with puromycin resistance. Two days later, 2.0 µg/mL puromycin antibiotic selection was added to the growth media. The remaining cells were then enriched by fluorescence-activated cell sorting (FACS) seven days after antibiotic selection. After three weeks in culture, the approximately 100 million cells grew, thereby generating a stable clone expressing the SERCA2a-PLB fusion biosensor at high levels. The stable cell line was maintained using F17 media (Sigma Alrdich, St. Louis, MO, USA) + (200 nM/mL) GlutaMAX + 2.0 µg/mL puromycin.

### 2.3. Homogenate Preparation

Cell lysates were generated for several cell-based assays. The stable cell lines were washed three times in phosphate buffer solution (PBS, with no magnesium or calcium added, Thermo Scientific, Waltham, MA, USA) by centrifugation at 300× *g*, and filtered using 70-µm cell strainers (Corning, Corning, NY, USA). Final resuspension was performed in homogenization buffer (0.5 µM MgCl_2_, 10 mM Tris-HCL ph 7.5, DNase I and protease inhibitor) at 6.65 × 10^6^ cells/mL, followed by incubation on ice for thirty min. Cells were then broken with the Tissumizer (Tekmar, Vernon, BC, Canada, SDT-1810) in 30-s bursts followed by a 5-min incubation on ice. This was repeated three times; homogenization was confirmed with a microscope. After homogenization, 2× sucrose buffer (1 mM MOPS, 500 mM sucrose, and protease inhibitor) was added for a final cell concentration of 13.3 × 10^6^ cells/mL.

### 2.4. Immunoblot Assay

Cell-homogenate samples were separated on a 4–20% polyacrylamide gradient gel (Bio-Rad, Hercules, CA, USA) loaded with 5 µg, 10 µg, and 15 µg in lanes, and then transferred to polyvinylidene fluoride (PVDF) membrane. This membrane was blocked using Odyssey Blocking Buffer with a two-hour incubation of primary mouse anti-SERCA2a (1:2500, Abcam, ab2861), followed by three 10 min TBST washes. Secondary IRDye680, goat anti-mouse antibody (1:5000) was incubated with the PVDF membrane for forty-five min at room temperature, followed by three 10-min TBST washes, then stored in Blocking Buffer. An Odyssey Scanner (LI-COR Biosciences, Lincoln, NE, USA) was used to image the immunoblots, and data were processed using LI-COR Image Studio™. 

### 2.5. NADH-Enzyme Coupled ATPase Activity Assay

Functional assays were performed using homogenate preparations expressing either the SERCA2a-PLB fusion biosensor or the appropriate donor-only control lacking PLB [7]. An enzyme-coupled, NADH-linked ATPase assay was used to measure SERCA2a Ca-ATPase activity in 96-well microplates. Each well contained assay mix (50 mM MOPS (pH 7.0), 100 mM KCl, 5 mM MgCl_2_, 1 mM EGTA, 0.2 mM NADH, 1 mM phosphoenol pyruvate, 10 IU/mL of pyruvate kinase, 10 IU/mL of lactate dehydrogenase, and 1 µM of the calcium ionophore A23187 from Sigma (St. Louis, MO, USA)), and added CaCl_2_ to set the free [Ca^2+^] to 10 µM [26]. 4 µg/mL of cell homogenate, CaCl_2_, compound, and assay mix were incubated for 20 min. The assay was started upon the addition of ATP, to a final concentration of 5 mM (200 µL total assay volume), and absorbance read at 340 nm in a SpectraMax Plus microplate spectrophotometer (Molecular Devices, Sunnyvale, CA, USA).

### 2.6. Calcium-Uptake Assays

Calcium uptake by pig cardiac SR vesicles was measured by adding 30 µg/mL of sample to a buffer containing 50 mM MOPS (pH 7.0), 100 mM KCl, 30 mg/mL sucrose, 1 mM EGTA, 10 mM KOA, 2 µM Fluo-4, and various amounts of CaCl_2_ calculated to reach the desired free [Ca^2+^]. The assay mix was dispensed into pre-plated 384-well drug plates and incubated at room temperature for 20 min while protected from light. A final concentration of 5 mM MgATP was added to start the reaction and the decrease in the fluorescence excited at 485 nm was monitored at 520 nm for 15 min using a FLIPR Tetra (Molecular Devices, San Jose, CA, USA).

### 2.7. HTS Cell Prep

On each day of HTS and FRET follow-up assays, approximately 100 million cells were harvested, washed three times in PBS with no magnesium or calcium Thermo Scientific (Waltham, MA, USA) by centrifugation at 300× *g*, filtered using 70-µm cell strainers Corning (Corning, NY, USA), and diluted to 1–2 million cells/mL using a Countess™ Automated cell counter (Invitrogen Carlsbad, CA, USA). On each day of screening, cell viability was assessed using the Trypan blue assay. After resuspension and dilution in PBS, the cells were continuously and gently stirred with a magnetic bar at room temperature to keep the cells in suspension and prevent clumping.

The library of pharmacologically active compound (LOPAC; 1280 small-molecules; Sigma (St. Louis, MO, USA) was initially formatted into 96-well mother plates using a Biomek FX liquid dispenser from Beckman Coulter (Brea, CA, USA). To create assay plates, LOPAC was subsequently re-formatted in 1536-well, black-wall/opaque-bottom assay plates at 50 nL (10 µM final concentration in the assay) using either an Echo^®^ liquid dispenser (Labcyte, San Jose, CA, USA) or a Mosquito liquid dispenser (TTP Labtech, Melbourn, UK). Control wells containing matching %v/v dimethylsulfoxide (DMSO) were formatted into unused wells and columns 1, 2, 47, and 48 of the assay plates.

Cells were dispensed into the 1536-well assay plates containing LOPAC compounds at room temperature, incubated for 20 and 120 min, and finally read in the FLT-PR using 2-channel detection. 

### 2.8. HTS Data Acquisition and Analysis after Data Acquisition

An in-depth description of our fluorescence instrumentation was previously published [7,9,10,20]. Briefly, this instrument uses a novel direct waveform recording (DWR) method, to acquire high-precision subnanosecond-resolved measurements of fluorescence lifetime from living cells in a 1536-well microplate within 3 min, measuring fluorescence lifetimes τ_D_ (donor only) and τ_DA_ (donor plus acceptor) from fluorescent proteins (typically ~2 ns) with SD of 10 ps (0.5%) or less [7]. This precision is 30 times better than achieved by measurements of fluorescence intensity and at least 100 times faster (higher throughput) than achievable with current commercial lifetime instruments, which acquire data with time-correlated single-photon counting (TCSPC, the same method used in FLIM). In the present study, modifications were made in the instrument to permit 2-channel detection, for the purpose of flagging false Hits due to interference from fluorescent compounds. This instrument and its performance are described below in the Results section (Figure 1). After data acquisition, each observed fluorescence waveform was convolved with the instrument response function to determine the mean lifetime τ (Equation (1)), and the energy transfer efficiency E was calculated from the mean lifetimes τ_D_ (donor only) and τ_DA_ (donor + acceptor) (Equation (2)*).* Z’ is a measurement of statistical significance. We utilized the robust Z’ (rZ’) instead of the standard Z’, so that extreme outliers do not excessively affect this assay quality gauge (Equations (3) and (4)). If rZ’ > 0.5, a Hit is considered robust (significant).
F_D_(t) = X_1_ exp(−t/τ_1_) + x_2_ exp(−t/τ_2_),τ = X_1_ τ_1_ + x_2_ τ_2_(1)
(2)$$FRET\;E=1-\frac{\tau_{\mathrm{DA}}}{\tau_{D}},$$
(3)$$rZ^{\prime }=1-\frac{3*\left(MAD_{Tg}+MAD_{DMSO}\right)}{\left|MedianFRET_{Tg}-Median\;FRET_{DMSO}\right|},$$
(4)MAD=1.4826∗ Median(|Xi−Median(X)|).

## 3. Results

### 3.1. Two-Channel Detection

For FLT-detected FRET-based HTS studies, we used a dual-channel high-throughput FLT-PR, derived from a single-channel model [7,9]. Figure 1A presents a diagram of the instrument, in which a single pulse from a high-energy microchip laser generates two separate subnanosecond-resolved FLT waveforms (Figure 1B) acquired at a rate of up to 1536 wells (one high-density microplate) in 2.5 min. A dichroic mirror splits the fluorescence signal into two channels (Ch1 and Ch2) at different emission wavelengths. Here, we demonstrate this 2-channel FLT detection in application to a GFP/RFP SERCA2a biosensor system.

Ch1 detects the FLT at the peak of the GFP emission spectrum (517 nm), while Ch2 detects the signal at 535 nm, where GFP emission is 43% of the peak signal (Figure 1B inset). A compound that changes the structure of the biosensor is likely to change FRET and thus the observed FLT in Ch1, with a negligible change in the ratio of Ch2 to Ch1 signal intensity. A significant change in this intensity ratio from 0.43 (> 3SD of the DMSO control average) is used as evidence of interference from fluorescence of the test compound. In many commercially available screening libraries, 5–10% of the compounds are fluorescent and are classified as pan-assay interference compounds (PAINS) [27]. A Hit is defined as a compound that changes the FLT signal by more than 3SD from the DMSO control. We found that the 2-channel detection system is effective for filtering out false Hits, i.e., those that change the FLT readout due to intrinsic fluorescence. This greatly decreases the number of false Hits, so that the small number remaining can be reliably ruled out later by lower-throughput assays, such as concentration-dependent FLT response and functional assays. We validated this rapid simultaneous filtering method by comparing it with data obtained from much slower separate detection of the full fluorescence emission spectrum [9,23].

### 3.2. SERCA2a-PLB FRET Biosensor

We developed a novel biosensor to detect FRET between human SERCA2a and PLB. The two proteins are tethered using a 47-residue linker sequence and expressed in HEK293 cells (Figure 2A). Fusion of PLB to the C-terminus of SERCA2a has been shown to preserve function, regulating SERCA2a via phosphorylation of PLB [24]. It has also been demonstrated that attachment of fluorescent proteins to the N-termini of PLB and SERCA2a does not interfere with PLB’s ability to bind and regulate SERCA2a [28]. We recently demonstrated that co-transfection of separately labeled GFP-SERCA2a (donor) or RFP-PLB (acceptor) produced sufficient FRET for detecting functionally relevant structural changes, from changes in the donor (GFP) FLT [13]. In that study, these two labeled proteins were co-transfected simultaneously into HEK293 cells. However, the variability of transient transfections posed several issues. It was observed that the transient co-transfections of GFP-SERCA2a and RFP-PLB provided several varying populations. The heterogeneous nature of transient transfection limited day-to-day screening reproducibility, as there are differences in the number of cells expressing either GFP-only and/or RFP-only, as demonstrated by flow cytometry analysis. The heterogeneity shows that expression of RFP-only cells (Q1) is similar to cells expression both RFP and GFP (Figure 2C). Therefore, in the present study, we concatenated (fused) on a single-polypeptide chain RFP-SERCA2a and GFP-PLB separated by a flexible 47-residue peptide linker. This construct consistently produces equimolar donor-acceptor (Figure 2C). A structural model of this concatenated SERCA2a-PLB biosensor indicates that the 47 amino acid linker provides adequate flexibility for GFP-PLB to productively bind to RFP-SERCA2a and be properly inserted into the membrane (Figure 2A).

With both components of the FRET biosensor tethered into a single expression vector, we were able to generate a stable cell line in which each donor-labeled PLB is always in close proximity to acceptor-labeled SERCA2a. This eliminated variability in expression level and the ratio of PLB to SERCA2a. HEK293-6E cells expressing the Epstein-Barr nuclear antigen 1 protein (EBNA) were transfected with the pTT22 carrying the cDNA corresponding to the RFP-SERCA2a-LINKER-GFP-PLB biosensor construct. Puromycin selection was added shortly after transfection, and the suspension cell population was enriched, approximately one week after transfection, through fluorescence activated cell sorting (FACS) isolation of cells expressing both GFP and RFP (Figure 2C). The cell line was then expanded to levels needed for HTS. Confocal microscopy (Figure 2B) confirmed that the biosensor is located within the endoplasmic reticulum and that GFP and RFP are consistently co-expressed. This is the first published demonstration of a stable cell line expressing a FRET biosensor in suspension-grown HEK293-6E cells.

By linking acceptor-labeled SERCA2a to donor-labeled PLB, we increased the likelihood of both FRET and SERCA2a-PLB functional interaction. In our previous work with an unfused SERCA2a-PLB biosensor, we attached the donor to SERCA2a [13], while we attached the donor to PLB in the present study. By positioning the RFP acceptor at the N-terminus, we minimized the impact of truncated expression or immature folding of the fusion biosensor, which would increase the donor-only population, thus reducing the magnitude of precision of FRET measurement. FACS analysis of the cell line expression the fusion biosensor (fusion cell line) shows a homogeneous population at the expected 1:1 ratio (Figure 2C, quadrant Q2), with only minor donor-only, acceptor-only, or untransfected cells (Figure 2C, quadrants Q3, Q1, and Q4, respectively). Monoclonal cell lines were generated through serial dilution [7]. The stable cells lines were shown to have a similar homogeneous population of cells expressing both GFP and RFP. (Appendix A). This same method was used to generate a donor-only cell line through the removal of the N-terminal RFP on SERCA2a. We used this cell line to determine the FRET efficiency of the fusion biosensor (Equation (1)).

### 3.3. SERCA2a-PLB Fusion FRET Biosensor Expression and Activity

Expression of labeled SERCA2a-PLB biosensors by the stable HEK293-6E cell line was confirmed by immunoblot (Figure 3A). Either the dual-labeled (donor + acceptor, DA) fusion, or single-labeled donor-only (D-only) control biosensor, was immunoblotted with anti-SERCA2a antibodies. In each case, we observed a single band at the predicted molecular weight of RFP-SERCA2a fused with GFP-PLB (Figure 3A, Appendix A). A lower molecular weight band around 100 kDa represents endogenous SERCA2b expression found in HEK293 cells. Three different sample concentrations were loaded for each sample.

Ca-ATPase assays on cell homogenates (Figure 3B) yielded a shift in the Ca^2+^-curves of the fusion construct containing PLB vs. the GFP-tagged SERCA2a-only control, as expected for PLB inhibition of SERCA2a. The control cell line, expressing only GFP fused to SERCA2a without PLB, has an apparent calcium affinity (pCa50) of 6.62 ± 0.04 (blue curve in Figure 3B), whereas SERCA2a directly fused to GFP-PLB shows a value of 6.38 ± 0.03 (red curve in Figure 3B). This pCa50 shift of 0.24 ± 0.07 is consistent with literature values for unlabeled proteins in cardiac SR [25,32,33]. We conclude that the FRET biosensor is functionally active and retains normal Ca^2+^-dependent inhibition by PLB.

### 3.4. HTS Results

The generation of a high-expressing, suspension-grown HEK293 stable cell line that detects structural changes between PLB and SERCA allows us to perform HTS to identify tool compounds. To validate the new protocol for flagging fluorescent compounds, and the performance of the fusion biosensor, we performed small-scale pilot screens of the LOPAC. Three repeats were performed. On each day of screening, suspensions of the stable fusion cell line were harvested and dispensed using a Multidrop™ Combi liquid dispenser (Thermo Scientific) into 1536-well assay plates that were preloaded with 1280 compounds and DMSO controls for a final 10 μM [compound]. Fluorescent compounds were flagged using the 2-channel FLT peak intensity ratio (as described above), and removed from subsequent follow-up analysis if they altered the 2-channel fluorescence intensity ratio beyond 3 SD of 0.1% vol/vol DMSO control wells. Each screen flagged 20–30 fluorescent compounds from the LOPAC library. These compounds were also evaluated using a spectral PR to confirm that they are indeed fluorescent compounds (FC). 27 compounds were flagged in one LOPAC screen as shown by red circles in Figure 4A. The majority of the fluorescent compounds were reproducibly flagged from screen to screen using a 3SD threshold defined as the percent difference of two-channel ratio, where the weakly fluorescent compounds were less reproducible from screen to screen. 

After removing fluorescent compounds, FRET Hits were assessed using the primary detection channel at 517 nm. Hits were selected based on a 5 or 7 SD threshold. The well-known SERCA inhibitor thapsigargin [34] was found to be a reproducible and robust FRET Hit from all three screens using the 7 SD threshold. The small molecule Ro 41-0960 was also found as a reproducible Hit using the 5SD threshold. This compound was previously identified as a SERCA2a effector using a related unfused, transiently-expressed SERCA2a-PLB biosensor [13]. The 23 reproducible Hits identified using 5 and 7 SD thresholds are shown as blue circles in Figure 4B. Raw data lifetime shown in Appendix A.

The reproducible Hits were further evaluated (triaged) by counter-screening against a mock biosensor construct consisting of GFP tethered to RFP via a 32-residue flexible linker (linker biosensor). This mock biosensor was used to flag promiscuous compounds that bind to the fluorescent proteins or produce other artifacts in the FLT readout that give the appearance of a change in FRET, and thus are falsely identified as Hits (false positives). After screening LOPAC multiple times with the linker biosensor, compounds that were found to alter FRET only from the fusion biosensor (large orange and green circles in Figure 4C). These compounds are shown as altering the FLT of the fusion biosensor by more than 5 SD but not the FLT of the null linker, using a more stringent threshold at 3SD (Figure 4C). Two of the FRET Hits have been previously demonstrated to alter SERCA2a’s function and are shown in Figure 4C as green circles. Compounds shown in orange did not show a significant functional effect. These results demonstrate the ability of the multichannel FLT-PR to eliminate interference from fluorescent compounds and reliably identify Hits that alter SERCA2a-PLB structure and function.

### 3.5. FRET and Functional Concentration-Response Curves (CRC)

The FRET Hits discovered at 10 µM using the SERCA2a-PLB fusion biosensor were cherry-picked, formatted, and dispensed in microplates for follow-up FRET and functional testing. Two of the five FRET Hits showed significant functional effects on SERCA2a. The subnanomolar inhibitor thapsigargin produced dose-dependent response from the fusion biosensor with an apparent FRET EC_50_ ≈ 7.5 nM thapsigargin, which matches the expected subnanomolar K_i_. The compound Ro 41-0960 produced a significant FRET change at micromolar concentrations; but due to its lower affinity, saturation was not observed (Figure 5A). Effects on the null linker were very slight and only observed at much higher concentration. 

To confirm functional specificity, we performed two additional assays, one to measure SERCA2a’s Ca-dependent ATP hydrolysis and one to measure SERCA2a’s transport of Ca^2+^ across the SR membrane. The absorbance-based NADH enzyme-linked ATPase assay was used to assess each compound’s ability to affect ATPase activity, at three different [Ca^2+^] (Figure 5B). Thapsigargin was found to completely inhibit SERCA2a at both high pCa 5.2 and mid-range pCa 6.2 calcium concentrations. SERCA activity was negligible at pCa8 and this basal activity was subtracted from high and mid-range calcium (Figure 5B). The previously identified SERCA2a-PLB effector Ro 41-0960 displays biphasic activation at pCa 6.2 (Figure 5B). At pCa 5.2, Ro 41-0960 inhibits SERCA2a ATPase activity. We seek compounds that activate SERCA2a at any pCa value, but we are especially interested in compounds that increase activity at pCa 6.2, where PLB has its greatest inhibitory effect. 

Oxalate-supported Ca^2+^-uptake assays were performed using an image-based instrument, the Molecular Devices FLIPR Tetra. Ca^2+^ uptake was measured as a change in Fluo-4 fluorescence. For these functional experiments, only the high (pCa 5.2) and mid-range (pCa 6.2) calcium concentrations were tested. Thapsigargin completely inhibited SERCA2a’s ability to pump Ca^2+^ at both pCa used. Ro 41-0960 increased SERCA2a’s ability to pump calcium at pCa 6.2 and inhibited at pCa 5.2 (Figure 5C). 

Ro 41-0960 is a catechol-O-methyltransferase (COMT) inhibitor [35]. Calcium transporter expression at the plasma membrane in the digestive system is decreased due to COMT inhibition [36]. Decreased expression of SERCA2a would decrease PLB’s interaction with SERCA2a, but this effect would take longer than the time the drug is incubated with the cells. A potential explanation is that Ro 41-0960 binds to the membrane-bound COMT enzyme and has off-target effects on PLB-SERCA2a interaction, causing indirect SERCA2a activation (e.g., by altering signal transduction pathways upstream of SERCA2a and PLB regulation, or altering gene expression regulators). These follow-up CRC experiments clearly show that the SERCA2a-PLB fusion biosensor detects structurally relevant compounds that alter SERCA2a’s function. A key parameter used to gauge the likelihood that a new biosensor can be successfully applied to a large-scale drug screen is the robust Z-factor (rZ’) (Equations (3) and (4)). A value of rZ’ between 0.5 and 1 indicates that the assay has at least 12 SDs of separation between the positive and negative controls. We utilized the robust Z’ instead of the standard Z’, so that extreme outliers do not affect the assay quality gauge [37]. We found the rZ’ of the SERCA2a-PLB fusion biosensor to be above 0.5 for three separate experiments, using a saturating dose of thapsigargin as a tool compound (Appendix A). These results demonstrate the feasibility of using this biosensor for a large-scale drug discovery campaign.

## 4. Discussion

These studies demonstrate the use of a novel multichannel FLT-PR capable of acquiring nanosecond FLT waveforms from fluorescent proteins expressed in live cells. In the past, we utilized a spectral HTS-PR to flag fluorescent compounds [9,23]. We are now able to record FLT waveforms simultaneously at two donor emission wavelengths and robustly flag fluorescent compounds, based on whether they differentially alter the signal at the two wavelengths (channels). A compound that alters the interaction of SERCA2a with its regulatory inhibitor PLB, changing only FRET, should affect both channels by the same factor. However, a fluorescent compound that has a different emission spectrum from the donor, as measured by the ratio of emission at 507 and 542 nm, should change the two channels by different factors, thus changing the ratio of the signals at the two channels. When compared to DMSO-containing control wells from these microplate screens, we found that fluorescent compounds are indeed flagged effectively using the two-channel methodology. We performed comparative screens on our spectral PR to validate this method and are currently performing larger screens to validate our Hit selection algorithms and technology.

Beyond the implementation of multichannel FLT acquisition, we also developed a single-polypeptide-chain biosensor to overcome the inherent limitations with co-transfecting two separate vectors to detect protein-protein interactions (Appendix A) [13]. We successfully generated stable cell lines using the EBNA-1 expressing HEK293 suspension-grown cells [38]. We found this new suspension-grown cell line to be more amenable (than adherent HEK293 cells) for scaling to large amounts. The pTT22 expression vector also allows for inducible or controlled protein expression, potentially needed for cytotoxic proteins. FACS isolation allowed for generation of stable cell lines and isolation of double-positive (GFP and RFP) cell lines. 

HTS of the LOPAC library with the SERCA2a-PLB fusion biosensor unambiguously identified a small-molecule, Ro 41-0960, that increases SERCA2a’s activity at a physiologically relevant calcium concentration. Through follow-up CRC testing, we confirmed that our FRET HTS assay can identify both inhibitors and activators of SERCA2a (Figure 5). The inhibitor thapsigargin (Tg) is known to bind SERCA2a tightly, locking it in the low calcium-affinity state known as E2 [39], which our previous FRET measurements showed is a low-FRET (open headpiece) structural state of SERCA2a [9]. For our SERCA2a-PLB biosensor, Tg slightly decreases FLT (increases FRET) (Figure 5A, left). Thus, the open E2 conformation of SERCA2a decreases the distance between the N-terminal domains of SERCA2a and PLB (Figure 2A). Ro 41-0960 enhances the Ca-transport activity of SERCA2a at intermediate Ca^2+^ (a desirable effect), while inhibiting at V_max_ Ca^2+^. However, this compound increases FRET, indicating that it acts by changing the conformation of the SERCA2a-PLB complex rather than by dissociating PLB from SERCA2a. This identifies Ro 41-0960 as a potential tool compound for HTS campaigns with large libraries of diverse and novel chemical matter. However, we do not expect Ro 41-0960 to be a useful SERCA2a-specific therapeutic due to its known action on other targets. 

## 5. Conclusions

We have shown that a catenated fluorescent biosensor, linking acceptor-labeled human cardiac SERCA2a to donor-labeled human cardiac PLB, enables increased sensitivity to interrogate protein-protein interactions that potently control Ca-transport in the myocyte. The biosensor retains regulatory function and is effective for HTS assays using an FLT-PR. In a small validation library (LOPAC), this biosensor identified two Hit compounds that alter SERCA2a-PLB interaction and SERCA2a activity. A second detection channel, at a different wavelength, efficiently filtered out false Hits caused by compound fluorescence. Our goal is to find small molecules that reduce the inhibitory interaction between PLB and SERCA2a, as needed for treatment of heart failure. We showed that Ca-uptake assays are essential for evaluating Hit effects on the SERCA2a activity that is directly relevant for myocyte function and thus for heart failure therapeutics—Ca-transport. Ca-ATPase assays are also important, as they inform the mechanism of action of a Hit, when used in tandem with the Ca-uptake assays. Future extension of this approach to perform large-scale HTS holds great promise for discovery of small-molecule activators of SERCA2a Ca-transport in the human heart, filling an urgent need in therapeutic development for heart failure. More generally, this approach, using catenated biosensors and FLT detection of FRET, should prove effective in mechanistic analysis and drug discovery for other systems involving functionally important protein-protein interactions.

## Figures and Tables

**Figure 1 cells-09-01170-f001:**
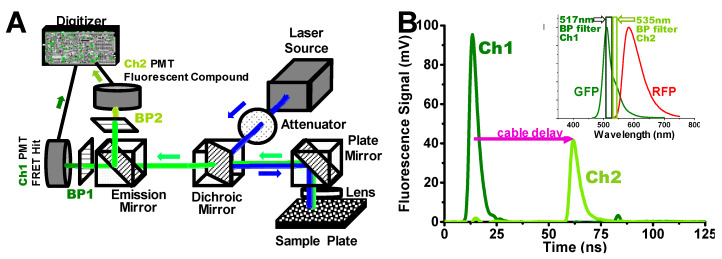
Dual-channel high-throughput fluorescence lifetime plate reader (FLT-PR), manufactured by Fluorescence Innovations Inc. and provided by Photonic Pharma, LLC. (**A**) This diagram is an overview of the instrument capable of detecting high-precision nanosecond-resolved FLT waveforms acquired at speeds of up to 1536 wells (one plate) in 2.5 min, which was originally based on single-channel detection [9]. Here a dichroic mirror splits the fluorescence signal into two, so it can be simultaneously detected by two separate photomultiplier tubes (PMT) at two wavelengths (channels). (**B**) The green and red fluorescent proteins (GFP) FLT signal at the peak of the GFP emission spectrum (inset), using a 517 ± 10 nm bandpass (BP) filter (Ch1). In the second channel (Ch2), detected simultaneously by a separate PMT, a 535 ± 7 nm BP filter (inset) is used to detect interference from fluorescent compounds (see text). The detection at Ch1 and Ch2 is simultaneous (from the same laser pulse), but the Ch2 signal is delayed by ~40 ns by a length of cable, to avoid electrical interference between the channels.

**Figure 2 cells-09-01170-f002:**
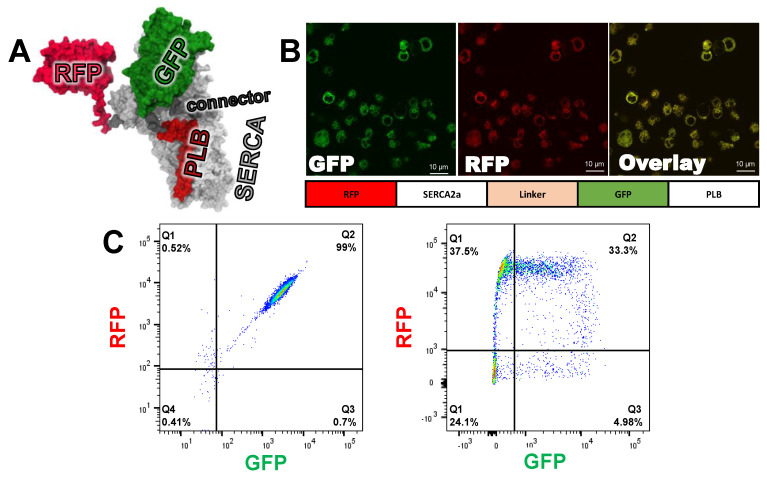
FRET biosensor engineering and characterization (**A**) Molecular model of concatenated sarcoplasmic reticulum Ca-pump (SERCA2a)- phospholamban (PLB) biosensor, based on crystal structures for SERCA2a-PLB (4KYT) [29], red fluorescent proteins (RFP) (3M22) [30], and GFP 1GFL) [31]. Modeling was done with Discovery Studio Visualizer (BIOVIA, San Diego, CA, USA). (**B**) Confocal microscopy of HEK293-6E suspension stable clone expressing the fusion biosensor at the ER membrane. (**C**) Representative flow cytometry of the stable fusion cell line expressing GFP and RFP at a 1:1 ratio, as shown by the 2D scatter plot (left), transient co-transfection of SERCA2a and PLB (right). The percentages of cells with GFP and/or RFP are indicated in each quadrant.

**Figure 3 cells-09-01170-f003:**
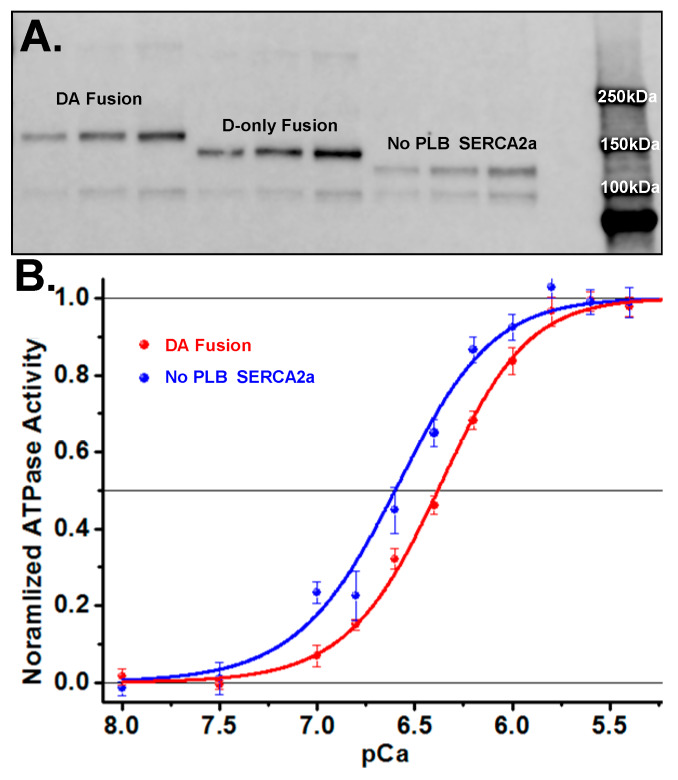
SERCA2a-PLB fusion FRET biosensor expression and activity (**A**) Western blot showing DA (GFP-SERCA2a-RFP-PLB) fusion biosensor (160 kDa) and D-only (GFP-SERCA2a, no PLB) fusion construct (137 kDa). (**B**) Ca^2+^-ATPase activity assays showing decreased apparent Ca^2+^ affinity for the SERCA2a-PLB fusion biosensor (blue, pCa50 = 6.62 ± 0.09) compared to the SERCA2a-only control (red, pCa50 = 6.38 ± 0.08). (SEM, n = 8.).

**Figure 4 cells-09-01170-f004:**
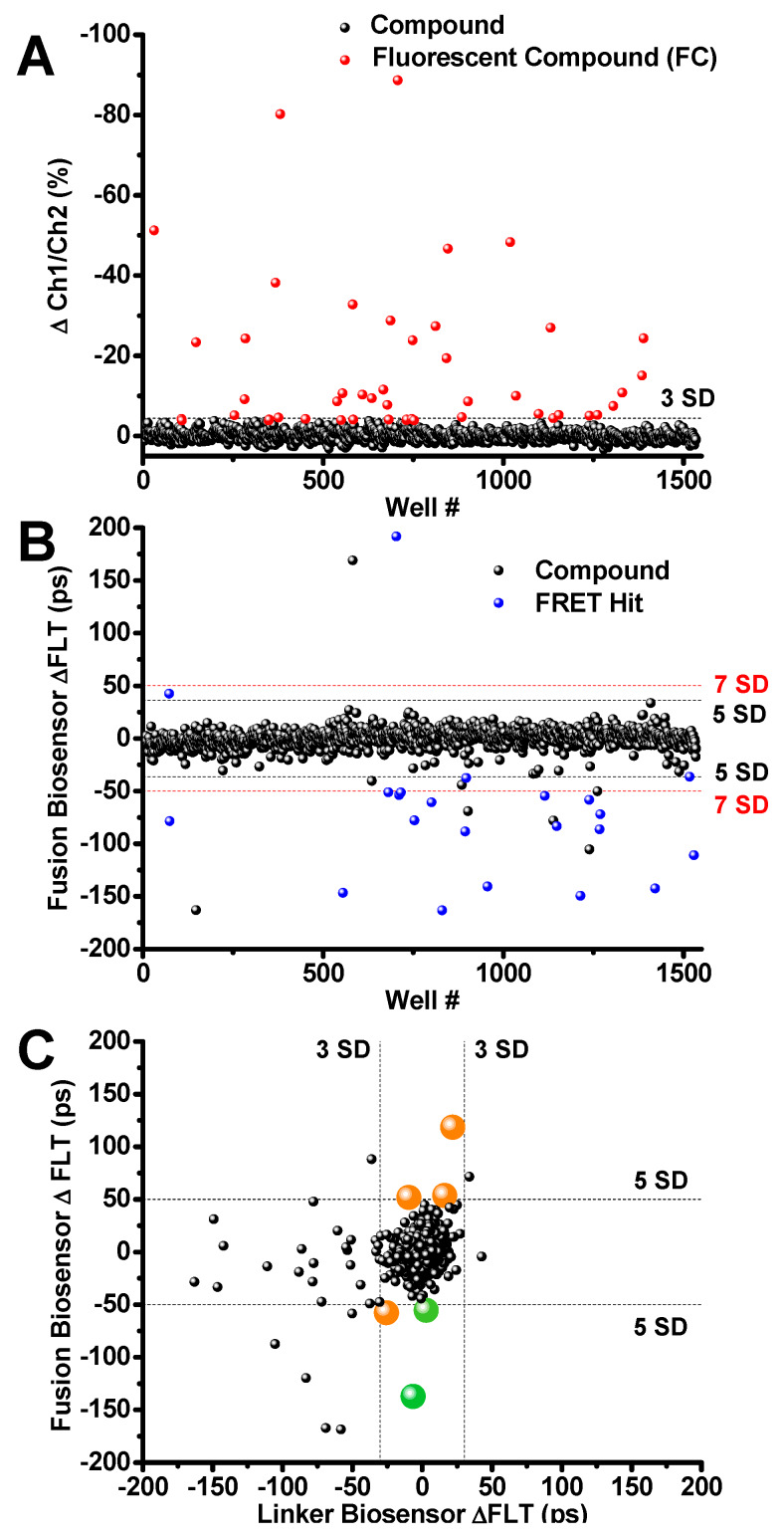
Library of pharmacologically active compound (LOPAC) Screen Results (**A**) Ch1/Ch2 intensity ratio flags 27 fluorescent compounds (FC; red). (**B**) Compounds that change the SERCA2a-PLB fusion biosensor FLT by ≥5SD were selected as Hits (blue). FCs were flagged and excluded from Hits. Two Hit thresholds are shown, 5SD and 7SD (black and red dotted lines, respectively. (**C**) Hits were further triaged by counter screening using the cell line expressing the GFP-RFP linker biosensor. Two reproducible Hits (thapsigargin and Ro 41-0960, shown in green) passed the linker biosensor test by changing its FLT by <3SD. Both compounds are known SERCA2a effectors. The remaining Hits are shown in orange. Raw lifetime screening data shown in Appendix A.

**Figure 5 cells-09-01170-f005:**
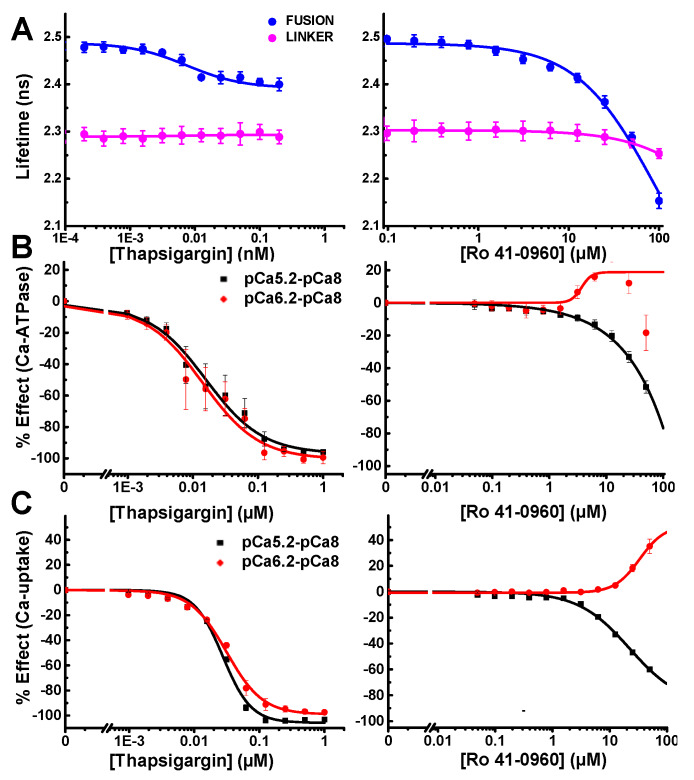
Concentration-response curve (CRC) analysis of LOPAC FRET Hits thapsigargin (left) and Ro 41-0960 (right). (**A**) FLT CRCs, using the fusion and linker biosensors, as indicated. (**B**) SERCA2a Ca-ATPase assays CRCs at [Ca^2+^] corresponding to maximal activity (pCa 5.2, black), intermediate (pCa 6.2, red). Basal activity (pCa8) was subtracted from each. (**C**) SERCA2a calcium-uptake monitored at pCa 6.2 and 5.2 using the Ca-sensitive dye Fluo-4 fluorescence. Each experiment was performed in duplicate (Ca-ATPase) or triplicate (Ca-uptake); error bars indicate SEM (n = 2 or 3). Curves generated using Hill fit.

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
