# Peer review of "Live-Cell Cardiac-Specific High-Throughput Screening Platform for Drug-Like Molecules That Enhance Ca2+ Transport"

_cells, 2020, doi:10.3390/cells9051170_

Round 1

Reviewer 1 Report

Please see summary and comments in the attached file.

Author Response

The authors demonstrate the development of an ongoing project which focuses on screening of small molecules that could affect the interaction between SERCA2a, which is essential for heart function, and PLB which is an inhibitor of SERCA2a. Similar screening tests have been reported previously by the group, and the current manuscript describes tests on a newly developed fusion protein between SERCA2a and PLB, with GFP and RFP (GFP-PLB ans RFP-SERCA). Using this FRET fusion protein they screen 1280 small molecules in living cells, and are able to exclude fluorescent compounds by using a two-detector setup. Hits are analyzed in a separate assay and shown to give the sought effect on Calcium uptake and Calcium dependent ATP hydrolysis.

The manuscript is very well written, and experiments and controls are well thought through and carried out. It deserves in my opinion to be published in Cells. A few minor suggestions for improvement are given below:

The abstract could clarify that the dual detector setup is not completely new for this manuscript, but something like an optimization of a system that has been used and reported on earlier. In Abstract and Introduction, it has now been clarified that the simultaneous dual detection is novel, specifically because of its application to HTS at extremely high speeds. (Abstract)

Row 96: “Second, we describe instrumentation changes that allow simultaneous detection of FLTs at two wavelengths..”. It is not clear to me whether the FLT of both detectors is being used in the analysis, please clarify this. It has now been clarified (last paragraph in Introduction) that the two detectors are simultaneously detecting signals at the two wavelengths, and this information is used in the analysis. (introduction, third paragraph)

Results, 3.1, Two-Channel Detection: Identification of a Fluorescent Compound (FC) is based on that FC has a different wavelength than EGFP (I assume that “GFP” refers to EGFP) and thus results in a shift in the intensity ratio between detectors 1 and 2, and at the same time the FLT (of detector 1?) does not change. However, if FC has a similar wavelength as EGFP, but a different FLT, then the intensity ratio will remain the same, and the measured FLT will be the mean of that from the FC and EGFP. EGFP has an intermediate FLT of 2.6 ns, so FCs may exist that have shorter as well as longer FLT. These will then be falsely be interpreted as changes in FRET. Could such compounds constitute some of the blue dots in figure 4b? Those with a negative delta FLT would then correspond to FCs with a FLT shorter than 2.6 ns. And, if this is the case, could such FCs be identified by the presence of a fluorescence lifetime decay with a dual decay time appearance (corresponding to the FLT of FC + EGFP)? Good point.  It is conceivable that some false hits will slip through this filter, but they are ruled out by subsequent lower-throughput assays such as concentration-dependent response.  This has been clarified. (Results 3.1, second paragraph)

Row 195: EGFP emission negligible at 535 nm? It is 43 % of max. Clarify also the window, above 535 nm? Or 525-545 nm? Please write out the numbers in the text. This has been corrected and clarified (Results 3.1, second paragraph).

Figure 4C: Two out of the seven were known since before and not analyzed further, if I understood it correctly. These two are marked in green in figure 4c. The remaining five could also be marked, in another color. We have implemented this suggestion in Figure 4.

Figure 5B: In the figure legend it says black=5.0, red=6.2, but in the text it says red=5.2, and that the other pCa level is 6.4. And the figure legend says it is biphasic at intermediate pCa (6.2, red curve) and that ro41-0960 inhibits completely at pCa 5.0, while the text says “At pCa 5.2 (red), Ro 41-0960 inhibits SERCA3a ATPase activity.” This has been corrected in figure and in text (Results 3.5 and Figure 5).

Row 364: nM instead of nm. This has been fixed (Results 3.5, first paragraph).

Row 387 (last paragraph of section 3.5): ”Decreased activity from calcium transporters..”. Does “calcium transporters” mean SERCA2a? Please clarify. Yes, this has been clarified. And “Decreased activity from calcium transporters expressed at the plasma membrane would decrease PLB’s interaction with SERCA, however…” Should it not be: “Decreased EXPRESSION of calcium transporters at the plasma membrane would decrease PLB’s interaction…”? Yes, this has been corrected (Results 3.5, last paragraph).

390: Please clarify what “off-target effects on PLB-SERCA interaction” means. This has been explained more clearly (Results 3.5, last paragraph).  

Reviewer 2 Report

The authors report on a High Throughput System (HTS) to probe intermolecular energy transfer between two cardiac proteins (PLB and SERCA2a) both linked to fluorescent proteins. It is a main goal to find out small molecules from a library that reduce the inhibitory interaction between PLB and SERCA2a, as needed for treatment of heart failure. Fluorescence lifetime (FLT) measurement in a 1536 well microtiter plate is a very valuable method for this purpose, and a scanning time of 2.5 minutes for a whole plate is well acceptable. But although FLT is the central technique of this paper, it is only scarcely described.

In Fig. 1 two photomultipliers for signal detection are reported. But how is this signal further processed and how can the time resolution be achieved? Which is the time resolution in comparison with the fluorescence lifetime of GFP? In this context it is worthwile to mention that deviations in the picosecond time range (see Figure 4) are used to characterize individual „hits“. Therefore, in a revised version of this manuscript lifetime data (not only deviations) and experimental resolution of lifetime measurements should be added.

Minor comments:

  • Abstract and further text throughout the manuscript: The term „Fluorescence resonance energy transfer“ is somehow misleading, since this energy transfer is non-radiative. For this reason several authors now use the term „Förster resonance energy transfer“, and the authors might consider whether this would be a better option.

  • Page 3, Figure 1: Bandwidths for spectral detection should be indicated. Furthermore, I cannot see how the curves depicted in this Figure can be used to resolve sub-nanosecond lifetimes. What is the reason for the time shift between the signals detected in the channels 1 and 2?

  • Are the calculations described on p.5 performed during or subsequent to the experiments?

  • Page 7 (1st part): Were all HTS experiments performed in cell suspensions?

Author Response

The authors report on a High Throughput System (HTS) to probe intermolecular energy transfer between two cardiac proteins (PLB and SERCA2a) both linked to fluorescent proteins. It is a main goal to find out small molecules from a library that reduce the inhibitory interaction between PLB and SERCA2a, as needed for treatment of heart failure. Fluorescence lifetime (FLT) measurement in a 1536 well microtiter plate is a very valuable method for this purpose, and a scanning time of 2.5 minutes for a whole plate is well acceptable. But although FLT is the central technique of this paper, it is only scarcely described.

In Fig. 1 two photomultipliers for signal detection are reported. But how is this signal further processed and how can the time resolution be achieved? This has been explained more clearly in the Figure 1 legend and in Section 3.1.

Which is the time resolution in comparison with the fluorescence lifetime of GFP? In this context it is worthwhile to mention that deviations in the picosecond time range (see Figure 4) are used to characterize individual „hits“. Therefore, in a revised version of this manuscript lifetime data (not only deviations) and experimental resolution of lifetime measurements should be added. It has now been explained more clearly (in citations to previous work and in Methods) that the precision of lifetime detection is better than 10 ps (Results 3.1, second paragraph). We have also added supplementary figure 6.

Minor comments:

  • Abstract and further text throughout the manuscript: The term „Fluorescence resonance energy transfer“ is somehow misleading, since this energy transfer is non-radiative. For this reason several authors now use the term „Förster resonance energy transfer“, and the authors might consider whether this would be a better option. FRET is detected by fluorescence, and “fluorescence resonance energy transfer” has been well-accepted terminology for decades by us and many other authors. In the second paragraph of Introduction, the revised ms now mentions that “Förster” is sometimes used (Introduction, second paragraph). 

 Page 3, Figure 1: Bandwidths for spectral detection should be indicated. Furthermore, I cannot see how the curves depicted in this Figure can be used to resolve sub-nanosecond lifetimes. What is the reason for the time shift between the signals detected in the channels 1 and 2? These issues have been clarified as indicated in the comment above. The bandwidth of the emission bandpass filters used acquiring the fluorescence emission of the nanosecond decay waveform have now been made to stand out for clarity (Results 3.1, second paragraph).

  • Are the calculations described on p.5 performed during or subsequent to the experiments? It has been clarified in Methods 2.8 that those calculations are performed after data acquisition (Methods 2.8).

  • Page 7 (1st part): Were all HTS experiments performed in cell suspensions? We have specified in text (Results 3.2, first paragraph).

Reviewer 3 Report

The manuscript deals with two things: (i) the cloning of a contatenated FP1-SERCA-PLB-FP2 construct (previously, a concatenated SERCA-PLB construct was available) for intramolecular FRET measurements of the SERCA-PLB interaction, (ii) the measurement of fluorescence decays from the FRET donor FP1 at two distinct spectral bands to be able to rule out false-positive readouts. While I applaud the usage of the fluorescence lifetime information for developing HTS assays, and in principle there is nothing wrong with the proposed methodology and validation thereof, I have some remarks that are related mostly to the way the methods are ‘sold’ to the reader. In essence, I ask to critically ‘tone down’ some parts of the paper, to allow the reader to more objectively judge the method.

Intro

“However, FLT (time-resolved) measurements offer major advantages over standard intensity (steady-state) measurements, since FLT captures an array of time-dependent intensity points in a single measurement with nanosecond resolution, in order to determine the change in FLT due to FRET [8, 9]. This results in a much greater information content and a 30-fold improvement in precision [7, 8].”

This is a misleading statement that needs to be rephrased. The authors refer to prior work which reported that intensity-based measurements of particular fluorescent samples exhibited a higher CV (coefficient of variation) because of concentration differences per sample, and other instrumental artifacts. Logically, lifetime measurements do not suffer from this. This, however, does not automatically render lifetime-based FRET also more precise; if FRET is quantified via sensitized acceptor emission, the precision and accuracy is exactly the same as when measured via fluorescence lifetimes. See f.e. Hellenkamp 2018 Nature Methods, although any publication where FRET is simultaneously quantified in the intensity and lifetime domain will report the same.

“Our FLT plate-reader enables these FLT measurements in HTS, dramatically increasing the precision, resolution, and accuracy of HTS assays.”

This is again a misleading statement that needs to be rephrased:

- Precision/accuracy, see above.

- ‘resolution’:  I’m assuming authors refer to time-resolution (state this explicitly), yet authors make it seem that this is an advantage. Again, in the context of this paper, where FRET is quantified, authors should specify exactly why the high time resolution of a TCSPC system is better to quantify FRET than the mere intensity.

Line 68: ‘dramatically increasing accuracy’: authors should explain why FLIM would be more accurate than intensity-based fluorescence in the context of what they measure: FRET.

Line 92: “This construct includes genetically encoded fluorophores on both SERCA and PLB, thus maximizing sensitivity to changes in the SERCA-PLB interaction.” Authors should specify why this maximizes sensitivity to the mentioned changes.

Results

Two and multicolor FLIM make me think immediately of methods such as nanosecond ALEX (kapanidis 2005 PNAS) and PIE (Mueller 2005 Biophys J), yet the authors do not separate donor and acceptor emission (which is quite useful for controlling the behavior of the FRET acceptor in the experiment; one can imagine a compound destroys the acceptor, which will unquench the donor, falsely making the experimenter think the compound is interaction-blocking), but merely look at two spectral bands of the FRET donor. I would think it’s fair to at least mention the advantages of alternating donor/acceptor excitation. There are some excellent review on PIE out there. I also have some questions regarding this point:

  • In figure 1, why is the blue decay temporally offset with respect to the black decay. There is no advantage to that, is there?
  • It would be easily possible (and much cheaper) to linearly unmix any background or unwanted signals from the FRET donor decay because the spectral signature of the FRET donor is known a priori. This should be discussed when referring to Figure 1 firstly, and elsewhere appropriate. In fact, I would have used the two detectors for specific donor and acceptor detection, to be able to quantify FRET both on the basis of lifetime as via sensitized acceptor emission. Combined with an extra acceptor exciting laser, one can then do PIE with all the advantages associated with that. This should be discussed.

Figure 2C (right): this flow cytometry signature looks absolutely strange. I see the following concentration relation between GFP and RFP:

  • Green varying, red low
  • Green varying, red high
  • Green low, red varying
  • Green high, red varying

Why is there no green varying, red varying relation, i.e. why are there no cells with both an intermediate green and intermediate red concentration?

Author Response

The manuscript deals with two things: (i) the cloning of a contatenated FP1-SERCA-PLB-FP2 construct (previously, a concatenated SERCA-PLB construct was available) for intramolecular FRET measurements of the SERCA-PLB interaction, (ii) the measurement of fluorescence decays from the FRET donor FP1 at two distinct spectral bands to be able to rule out false-positive readouts. While I applaud the usage of the fluorescence lifetime information for developing HTS assays, and in principle there is nothing wrong with the proposed methodology and validation thereof, I have some remarks that are related mostly to the way the methods are ‘sold’ to the reader. In essence, I ask to critically ‘tone down’ some parts of the paper, to allow the reader to more objectively judge the method.

Intro

“However, FLT (time-resolved) measurements offer major advantages over standard intensity (steady-state) measurements, since FLT captures an array of time-dependent intensity points in a single measurement with nanosecond resolution, in order to determine the change in FLT due to FRET [8, 9]. This results in a much greater information content and a 30-fold improvement in precision [7, 8].”

This is a misleading statement that needs to be rephrased. The authors refer to prior work which reported that intensity-based measurements of particular fluorescent samples exhibited a higher CV (coefficient of variation) because of concentration differences per sample, and other instrumental artifacts. Logically, lifetime measurements do not suffer from this. This, however, does not automatically render lifetime-based FRET also more precise; if FRET is quantified via sensitized acceptor emission, the precision and accuracy is exactly the same as when measured via fluorescence lifetimes. See f.e. Hellenkamp 2018 Nature Methods, although any publication where FRET is simultaneously quantified in the intensity and lifetime domain will report the same. The subject of this paper is FRET measurements for high--throughput screening (HTS).  The Hellenkamp manuscript describes single-molecule FRET, which is not conducive to HTS. This is now noted in our paper (Introduction, second paragraph).

 “Our FLT plate-reader enables these FLT measurements in HTS, dramatically increasing the precision, resolution, and accuracy of HTS assays.” This is again a misleading statement that needs to be rephrased: We have rephrased this and this technical advance strictly applies to HTS. See next comment (introduction, second paragraph)

- Precision/accuracy, see above. We have clarified this (Introduction, second paragraph).

- ‘resolution’:  I’m assuming authors refer to time-resolution (state this explicitly), yet authors make it seem that this is an advantage. Again, in the context of this paper, where FRET is quantified, authors should specify exactly why the high time resolution of a TCSPC system is better to quantify FRET than the mere intensity. We are not using TCSPC. We are using direct waveform recording (DWR) and have cited the appropriate references which explain the difference. We are describing the precision increase to relation to TCSPC (Introduction, second paragraph)

Line 68: ‘dramatically increasing accuracy’: authors should explain why FLIM would be more accurate than intensity-based fluorescence in the context of what they measure: FRET. “accuracy” has been deleted. Thank you for your comment (Introduction, second paragraph).

Line 92: “This construct includes genetically encoded fluorophores on both SERCA and PLB, thus maximizing sensitivity to changes in the SERCA-PLB interaction.” Authors should specify why this maximizes sensitivity to the mentioned changes.  It is now explained more clearly that, by fixing the donor-acceptor ratio at 1, we greatly decrease the range of FRET values for the  control (no compound), while greatly improving day-to-day reproducibility (Introduction, Thrid paragraph).

Results

Two and multicolor FLIM make me think immediately of methods such as nanosecond ALEX (kapanidis 2005 PNAS) and PIE (Mueller 2005 Biophys J), yet the authors do not separate donor and acceptor emission (which is quite useful for controlling the behavior of the FRET acceptor in the experiment; one can imagine a compound destroys the acceptor, which will unquench the donor, falsely making the experimenter think the compound is interaction-blocking), but merely look at two spectral bands of the FRET donor. I would think it’s fair to at least mention the advantages of alternating donor/acceptor excitation. There are some excellent review on PIE out there. I also have some questions regarding this point: This has been addressed. Previous single molecule FRET, FLIM, and PIE based measurements have been able to achieve great precision however, their application is not conducive to high throughput based methods(HELLENKAMP)(Kapanidis 2005 PNAS) (Mueller 2005 Biophys J.) (Introduction, second paragraph).

  • In figure 1, why is the blue decay temporally offset with respect to the black decay. There is no advantage to that, is there? This has been explained more clearly in the Figure legend 1. Each channel has its own photomultiplier tube (PMT) detector for the high speed fluorescence lifetime measurement. The PMT voltage is adjusted to optimize the signal intensity in each channel. Each PMT channel is connected to the digitizer through a variable length of coaxial cable, in order to delay the signal (~1ns per 30cm), so that each signal is resolved from the laser pulse and the other single, in order to avoid electronic interference. (Figure legend 1)

It would be easily possible (and much cheaper) to linearly unmix any background or unwanted signals from the FRET donor decay because the spectral signature of the FRET donor is known a priori. This should be discussed when referring to Figure 1 firstly, and elsewhere appropriate. In fact, I would have used the two detectors for specific donor and acceptor detection, to be able to quantify FRET both on the basis of lifetime as via sensitized acceptor emission. Combined with an extra acceptor exciting laser, one can then do PIE with all the advantages associated with that. This should be discussed. This has been addressed. Previous single molecule FRET, FLIM, and PIE based measurements have been able to achieve great precision however, their application is not conducive to high throughput based methods(HELLENKAMP)(Kapanidis 2005 PNAS) (Mueller 2005 Biophys J.) (Introduction, second paragraph).  

Figure 2C (right): this flow cytometry signature looks absolutely strange. I see the following concentration relation between GFP and RFP:

  • Green varying, red low
  • Green varying, red high
  • Green low, red varying
  • Green high, red varying

Why is there no green varying, red varying relation, i.e. why are there no cells with both an intermediate green and intermediate red concentration? This plot depict the nonhomogenous nature of transient transfections of two separate constructs. The heterogeneity is shown because expression of RFP only cells is similar to double positive, This figure show the difference from the fused biosensor compared to the previous transient unfused biosensor. (Results, 3.2, First paragraph).